Eukaryotic gut community of the bat Myotis arescens in anthropized landscapes in Chile

Ramírez-Fernández Lia 1 2
Saldarriaga-Córdoba Mónica 3 4
http://orcid.org/0000-0002-4346-5524 Silva Andrea X. 5
http://orcid.org/0000-0002-7081-6975 Napolitano Constanza 6 7 8
Rodríguez-San Pedro Annia 9 ar.sanpedro@gmail.com
1 Laboratorio de Genómica de Ambientes Extremos, Facultad de Recursos Naturales Renovables, Universidad Arturo Prat , Iquique , Chile
2 Núcleo de Investigación Aplicada e Innovación en Ciencias Biológicas, Facultad de Recursos Naturales Renovables, Universidad Aturo Prat , Iquique , Chile
3 Centro de Investigación en Recursos Naturales y Sustentabilidad (CIRENYS), Universidad Bernardo O’Higgins , Santiago , Chile
4 Escuela de Medicina Veterinaria, Universidad Bernardo O’Higgins , Santiago , Chile
5 AUSTRAL-omics, Vicerrectoría de Investigación, Desarrollo y Creación Artística, Universidad Austral de Chile , Valdivia , Chile
6 Departamento de Ciencias Biológicas y Biodiversidad, Universidad de los Lagos , Osorno , Chile
7 Institute of Ecology and Biodiversity , Concepción , Chile
8 Cape Horn International Center , Puerto Williams , Chile
9 Centro de Investigación e Innovación para el Cambio Climático (CiiCC), Facultad de Ciencias, Universidad Santo Tomás , Santiago , Chile
Kormas Konstantinos
Electronic publication date: 2025 Jun 30
Publication date: 2025
Volume: 13
Electronic Location ID: e19563
Received 2025 Feb 2; Accepted 2025 May 13
Copyright: © 2025 Ramírez-Fernández et al.
Copyright year: 2025
Copyright holder: Ramírez-Fernández et al.
License: This is an open access article distributed under the terms of the Creative Commons Attribution License, which permits unrestricted use, distribution, reproduction and adaptation in any medium and for any purpose provided that it is properly attributed. For attribution, the original author(s), title, publication source (PeerJ) and either DOI or URL of the article must be cited.
License URL: https://creativecommons.org/licenses/by/4.0/

Keywords: Anthropization, Bat gut microbiome, Myotis arescens, Human–bat interactions, Native vegetation, Zoonotic pathogens, Eukaryotic microbes, Public health risks, Habitat fragmentation

Funding: Chilean National Agency for Research and Development (ANID, Chile) Fondecyt Regular 1220758-1251063 Fondecyt de Iniciación 11240050-11250198 ANID PIA/BASAL FB210006 ANID/BASAL FB210018 Vicerrectoría de Investigación y Postgrado, Universidad de Los Lagos This work was funded by the Chilean National Agency for Research and Development (ANID, Chile), ANID Fondecyt Regular 1220758-1251063, Fondecyt de Iniciación 11240050-11250198, ANID PIA/BASAL FB210006, ANID/BASAL FB210018. Vicerrectoría de Investigación y Postgrado, Universidad de Los Lagos funded the APC for the article. The funders had no role in study design, data collection and analysis, decision to publish, or preparation of the manuscript.

==============================
Background

Human-driven environmental changes can disrupt wildlife habitats, forcing animals to adapt to fragmented or degraded ecosystems. In some cases, this leads to increased proximity between wildlife and human populations, heightening the risk of pathogen spillover. Bats, as key ecological players, are particularly sensitive to such disturbances. While some species decline in heavily altered environments, others adapt and thrive near human settlements, increasing the likelihood of encounters. Given that bats can host a wide range of zoonotic pathogens, this adaptive behavior raises important public health concerns. Despite their ecological significance and their role in zoonotic disease dynamics, the gut eukaryotes communities associated with bats remain less studied.

Methods

This study focused on the Valparaíso Myotis (Myotis arescens), an insectivorous bat species endemic to central Chile that is significantly impacted by anthropogenic deforestation and habitat fragmentation. We characterized the gut eukaryotic communities of M. arescens through fecal sample analysis. Targeted microbial groups included fungi, metazoan parasites, and protists. High-throughput sequencing was employed to assess gut eukaryotes diversity, and beta diversity analysis was conducted to explore clustering patterns in relation to environmental variables, such as vegetation cover and land use types.

Results

Our analyses revealed that the gut eukaryotic community of M. arescens consistently included taxa from the Apicomplexa, Ascomycota, and Basidiomycota phyla, with Apicomplexa being the most abundant. Beta diversity analysis showed distinct clustering by sampling location, with the percentage of native vegetation identified as the primary factor shaping gut eukaryotic community structure. Other influential variables included the presence of annual crops, orchards, water bodies, and urban areas. Notably, a high abundance of Apicomplexa—particularly amplicon sequence variants (ASVs) related to the genus Eimeria—was detected in bat feces across sites with varying degrees of anthropogenic disturbance.

Conclusions

This study highlights the significant role of native vegetation in shaping the eukaryotic gut community of M. arescens, suggesting that gut eukaryotic composition can serve as a bioindicator of bat health and habitat quality. Among the dominant taxa, members of the genus Eimeria were frequently detected across sites with varying degrees of anthropogenic disturbance. Although Eimeria is generally considered host-specific and not zoonotic, its high prevalence in bat gut communities points to the need for further research into its ecological role and potential implications for wildlife health. Overall, these findings underscore the importance of conserving native habitats to maintain ecosystem integrity and support healthy bat populations.

Introduction

Land use change refers to the process of human activities transforming natural landscapes into human-modified environments, such as urban areas, agriculture land, and infrastructure (Lambin, Geist & Lepers, 2003). This transformation can have significant impacts on pristine and wild environments, including changes in habitat structure (Zhao et al., 2025), biodiversity (Newbold, 2018; Davison, Rahbek & Morueta-Holme, 2021), and ecosystem functioning (Gomes et al., 2021). One aspect of land use change that has gained attention is its potential influence on the interactions between wildlife and zoonotic diseases (Rubio, Fredes & Simonetti, 2016; White & Razgour, 2020; Plowright et al., 2021). Anthropization often involves habitat destruction and fragmentation (Fahrig, 2003), forcing animals to adapt to new, patchy habitats, which could alter their behavior, movement patterns, and resting sites (Meyer, Struebig & Willig, 2016). These changes in behavior and habitat use could influence their contact with other animals and their potential exposure to pathogens (Loh et al., 2022; Gibb et al., 2020; Plowright et al., 2021).

Bats are flying mammals that play a key role in the world’s ecosystems, acting as seed dispersers (Aziz et al., 2021; da Silva et al., 2024), pollinators (Ramírez-Fráncel et al., 2022), insects controllers (Tuneu-Corral et al., 2023), and nutrient recyclers (Lundberg, McFarlane & Van Rentergem, 2022). Bats are also known to host of a wide range of pathogens, including many with zoonotic potential (Mühldorfer, 2013; Brook & Dobson, 2015; Federici et al., 2022; Barbosa et al., 2023). Replacing natural habitats for agriculture or urban development can increase interactions between humans and bats, leading to more opportunities for zoonotic disease transmission (Frick, Kingston & Flanders, 2020). While some spillover events occur indirectly—such as Hendra virus being transmitted from fruit bats to horses and then to humans (Eby et al., 2022)—habitat disturbance can also facilitate direct contact, for example through contamination of water sources with bat feces from coronavirus-positive bats (Meyer et al., 2024) or through the consumption of bushmeat from Ebola-infected bats (Leroy et al., 2009). Furthermore, it has been shown that zoonotic host diversity tends to increase in human-dominated ecosystems, potentially enhancing the risk of pathogen spillover events (Gibb et al., 2020). The main zoonotic pathogens for which bats act as natural reservoirs are viruses, many of which have caused outbreaks worldwide (Allocati et al., 2016; Letko et al., 2020). Although research on eukaryotic pathogens in bats and their potential role in human diseases is emerging (Barbosa et al., 2023; Corduneanu et al., 2023; Karunarathna et al., 2023), it remains comparatively scarce relative to the vast literature on viral and bacterial pathogens. Given the potential risk associated with zoonotic disease spillover, it is important for researchers, conservationists, and policymakers to study and manage human-bat interactions, particularly in regions where few data exist. This involves understanding the ecology of bats, the pathogens infecting them, and the ways in which human activities intersect with their habitats. This information is crucial for preventing future disease outbreaks and maintaining the delicate balance of ecosystems.

In Chile, where 17 bat species have been recorded, including Myotis and Lasiurus genus (D’Elía et al., 2020; Rodríguez-San Pedro et al., 2022, 2023; Novaes et al., 2022), and despite the vital ecological roles that bats play—such as insect control and pollination—few studies have focused on understanding their associated pathogens. It is important to communicate bat-borne diseases with caution, as bats do not pose immediate public health risks unless their habitats are disturbed (Plowright et al., 2024; Weber et al., 2023). The Valparaiso myotis (Myotis arescens) is a small insectivorous bat endemic to central Chile. It is a member of the vesper bat family (Vespertilionidae) and is one of the smallest Myotis species found in the country, with a forearm length of 34–37 mm and a body mass of 5–7 g (Novaes et al., 2022). It forages primarily in open areas, woodlands, and near water sources, using echolocation to hunt for insects such as moths, flies, and beetles, which helps reduce pest populations (Rodríguez-San Pedro et al., 2020). Central Chile has experienced extensive land-use change, leading to significant loss and fragmentation of the native Mediterranean sclerophyllous forest and shrubland (Echeverria et al., 2006; Zamorano-Elgueta et al., 2015; Miranda et al., 2017), which may affect bat habitats and influence pathogen dynamics.

In this study, we characterized the eukaryotic gut community of M. arescens. This microeukaryote community includes nucleated organisms such as fungi (filamentous fungi and yeasts), metazoan parasites (cestodes, nematodes, and helminths) and protists (Laforest-Lapointe & Arrieta, 2018). The vast majority of microbiome studies are limited to the bacterial component of this symbiotic community (Ingala et al., 2019; Lobato-Bailón et al., 2023). However, despite being described as playing important ecological roles in the microbiome as well as in host health, the fraction of eukaryotic gut community has been largely overlooked in animal microbiome studies compared with its prokaryotic counterpart (Vargas-Albores et al., 2023). Here, we assess bat colonies dwelling in landscapes with different levels of land use, focusing on the potentially pathogenic species detected in the bat guano. With increasing urbanization and land-use changes, this study aims to understand how human activities may impact the habitats and populations of this endemic species, as well as the potential risks associated with human-bat interactions in terms of zoonotic disease transmission.

Materials and Methods

Sampling sites and bat fecal collection

The study areas are located in the Metropolitan and O’Higgins regions of central Chile. These regions are characterized by agricultural landscapes composed of a variety of crops, including horticultural species, cereals, alfalfa fields, orchards, vineyards, in addition to small remnants of native vegetation (sclerophyllous forest or shrublands), exotic tree plantations (Pinus sp. or Eucalyptus spp.), and urban or semiurban areas. According to the Köppen climate classification, the region has a temperate pluviseasonal Mediterranean climate (Peel, Finlayson & McMahon, 2007), with most rainfall concentrated in the winter season, June–August. The mean annual precipitation is approximately 360 mm, and the mean annual temperature of 15 °C (Luebert & Pliscoff, 2017). Fieldwork was approved by the National Agency for Research and Development, ANID (formerly CONICYT), under the sponsorship of the University of Chile (project number: 3160188).

To assess the gut eukaryotic community composition, we analyzed bat fecal samples, a widely validated proxy in microbiome studies (Thomas, Clark & Doré, 2015). Samples were collected from four M. arescens colonies, located in separated landscapes. Each colony inhabited human-made structures as roost sites, with colony sizes ranging from 40–100 individuals. These landscapes varied in their degrees of anthropogenic disturbance, defined as the percentage of native vegetation remaining within a 5 km radius of each colony, along with land cover or use types (Fig. 1). This buffer area was established considering documented foraging ranges of comparable insectivorous bat species with similar body sizes to those present in central Chile. Land cover classification was performed through manual digitization of 2017 Google Earth imagery at a 1:15,000 scale using QGIS software. Ten habitat categories were identified: riparian habitats, annual crops, orchards, water bodies, prairies, vineyards, rural areas, urban areas, hedgerows (linear vegetation barriers between fields), and native vegetation comprising sclerophyll forests and shrublands. This classification covered the dominant vegetation types in the study area. Spatial analysis was conducted using Patch Analyst 5.2 extension in ArcGIS 10.4.1. Detailed characteristics of the sampled landscapes are provided in Table 1.

Figure 1 Study area and sampling sites.

Map of the study area showing (A) the location of the four sampling sites (black stars), (B) referential images of the landscape surrounding sampling sites: site 1 (upper left), site 2 (upper right), site 4 (lower left), and one of bat’s colonies (lower right).

Table 1 Land use and vegetation cover surrounding sampling sites.

Sampling sites	Coordinates	Rip (%)	NaV (%)	AnC (%)	Orch (%)	Wat (%)	Pra (%)	Ru (%)	Urb (%)	Hed (%)	Vin (%)	
Site 1	33°51′57.61″S 70°35′21.10″W	0.76	77.63	0.62	6.05	0.33	0.00	0.63	1.57	4.56	0.00	
Site 2	33°42′16.77″S 70°40′28.23″W	3.82	31.94	10.74	23.21	0.23	0.05	0.77	5.06	13.19	0.23	
Site 3	33°53′46.18″S 71°13′58.29″W	0.00	67.42	5.95	4.79	0.17	0.29	0.49	14.24	1.17	0.32	
Site 4	34°36′31.31″S 71°7′40.51″W	7.04	48.54	10.60	4.42	0.02	0.26	0.34	5.72	2.10	0.13	
Note:

Proportion of land uses and vegetation cover types that surround the sites where the colonies under study are located. Rip, Riparian habitats; NaV, native vegetation; AnC, annual crops; Orch, orchards; Wat: water bodies; Pra, prairie; Ru: rural areas; Urb, urban areas; Hed, hedgerows; Vin, vineyards fields.

Each colony was visited in March 2017. For each colony, three pooled fecal samples were collected, each pooled sample containing an average of 20 pellets. Feces were collected directly from the roost floor using sterile forceps. To ensure the freshness of bat fecal samples, we cleared the collection areas prior to the sampling period. This allowed us to confirm that any droppings found afterward were recently deposited. Samples were collected early in the morning, immediately following the bats’ overnight activity, further increasing the likelihood of obtaining fresh material. Only fresh samples were collected and all were stored in sterile Eppendorf tubes and kept at −20 °C (within 10 h of collection) to preserve the DNA prior to analysis (Hale et al., 2016).

DNA extraction and library preparation

DNA was extracted using the PowerFecal® DNA Isolation Kit (MoBio Laboratories, Carlsbad, CA, USA) following the manufacturer’s instructions, with the exception that the DNA was eluted in 25 μL of Milli-Q water.

To amplify one portion of the 18S rRNA gene, the primer combination 7f forward and 570r reverse was used (Noyce et al., 2016). Library preparation followed the Illumina 16S Metagenomics Sequencing protocol, adapted for 18S rRNA gene. In addition to the specific primers, the sequence was supplemented with the design outlined by Fadrosh et al. (2014), which includes a linker sequence, an index sequence, and a heterogeneity spacer. Amplicons were sequenced using 600-cycle kits with paired-end technology on an Illumina MiSeq sequencer (Illumina, San Diego, CA, USA). DNA extraction, library preparation, and next-generation sequencing were carried out at AUSTRAL-omics core research facility of the Universidad Austral de Chile (Valdivia, Chile). The 18S metabarcoding dataset has been submitted to the European Nucleotide Archive (ENA) under the accession number PRJEB78431.

Bioinformatic and statistics analysis

Raw FASTQ files from 18S rRNA sequencing were processed using the DADA2 pipeline (Callahan et al., 2016), accessible at https://benjjneb.github.io/dada2/tutorial.html in R (Oksanen et al., 2013). This software performs fast and accurate inference of amplicon sequence variant (ASVs) with a single-nucleotide resolution, improving the precision of community composition analyses in metabarcoding studies.

Forward and reverse reads were truncated at 250 bp using the truncLen parameter. Sequences containing ambiguous bases (maxN > 0) or with expected errors exceeding 8 (maxEE = 8) were removed. A minimum quality threshold of Q = 0 (truncQ = 0) was applied, and PhiX control sequences were filtered out (rm.phix = TRUE). Filtered reads were used for ASV inference, and chimeric sequences were identified and removed using the removeBimeraDenovo() function with the consensus method.

Taxonomy assignment was performed using the SILVA v.132 database (Quast et al., 2013). ASVs assigned to the phyla Vertebrata, Arthropoda, or Phragmoplastophyta phyla were excluded from subsequent analyses. For alpha diversity analysis, the Shannon, Observed Richness and Simpson indices were calculated using the diversity function from the vegan package in R (Oksanen et al., 2013). To test for differences between sites, we first applied Kruskal-Wallis tests after assessing the data for normality using the Shapiro-Wilk test. In addition, we fitted generalized linear models (GLMs) to each diversity metric: a quasi-Poisson GLM was used for Observed richness to correct for overdispersion, and Gaussian GLMs were used for Shannon and Simpson indices. Rarefaction curves were generated using the ggrare function from the microbiomeSeq package in R. This function plots species richness (observed ASVs) as a function of sequencing depth across samples. Confidence intervals were displayed using standard error (se = TRUE), and samples were color-coded by sampling site.

To evaluate beta diversity and patterns of community composition across sampling sites, we performed non-metric multidimensional scaling (NMDS) analyses using two dissimilarity indices: Bray–Curtis (which considers relative abundance) and Jaccard (based on presence–absence data). For each index, we computed a dissimilarity matrix and performed NMDS using the metaMDS() function from the vegan package in R, with two dimensions (k = 2) and 100 random starts (trymax = 100) to ensure convergence on a stable solution. Stress values were used to assess the goodness of fit.

To identify environmental variables significantly associated with the NMDS ordinations, we applied the envfit() function from vegan, which fits environmental vectors onto the ordination space and uses permutation tests (n = 999) to assess significance. This analysis was applied to both Bray–Curtis and Jaccard ordinations.

Differences in community composition between sites were tested using permutational multivariate analysis of variance (PERMANOVA) via the adonis2() function, also from vegan, using both Bray–Curtis and Jaccard dissimilarity matrices. We used 999 permutations and evaluated the proportion of variance explained by the factor Site in each case.

The environmental matrix included site-level landscape characteristics recorded at each sampling location: riparian vegetation (Rip), natural vegetation (NaV), anthropogenic cover (AnC), orchards (Orch), water bodies (Wat), pastures (Pra), rural areas (Ru), urban cover (Urb), hedges (Hed), and vineyards (Vin). These variables were tested for correlations with gut eukaryotic community structure.

To summarize environmental gradients and reduce collinearity among landscape variables, we performed a principal component analysis (PCA). The first two principal components (PC1 and PC2) were used to visualize site separation and interpret dominant environmental gradients. These components were incorporated into vector fitting analyses (envfit), PERMANOVA models, and generalized linear models to assess their association with gut eukaryotic community diversity, using the vegan and stats packages in R.

A heatmap of relative abundance percentages by site was constructed using amp_heatmap, and boxplots were created with amp_boxplot, both from the ampvis2 package in R.

Phylogenetic analysis of Apicomplexa ASVs

Amplicon sequence variants (ASVs) assigned to the phylum Apicomplexa were extracted from the phyloseq object. Sequences were aligned with the AlignSeqs() function from the DECIPHER package, and the alignment was converted to a phyDat object. A preliminary phylogenetic tree was inferred using the Neighbor-Joining method and optimized under the GTR + I + Γ model with the phangorn package. The final tree was visualized with ggtree, and ASVs were labeled with genus-level taxonomic annotations. To illustrate ASV distribution across sampling sites, proportional colored circles were added to each tip, representing relative abundance and site of detection.

Results

Raw data analysis and gut eukaryotic diversity

A total of 71,995 raw Illumina sequence reads were obtained from 12 samples. After quality filtering and chimera removal, 58,585 high-quality sequences remained. To minimize potential contamination from host or diet-derived DNA, ASVs assigned to the phyla Vertebrata, Arthropoda, and Phragmoplastophyta were excluded. The final dataset used for downstream diversity analyses consisted of 37,064 sequences distributed across 211 ASVs.

Rarefaction analysis revealed that samples of Site 2 and Site 3 harbored the highest eukaryotic richness, requiring greater sequencing effort to capture their diversity. In contrast, Site 1 and Site 4 showed lower richness and reached saturation with fewer sequences. The plateauing of all curves indicates that sequencing depth was generally sufficient to capture species richness across sites (Fig. S1).

No significant differences in alpha diversity were detected among sites based on Kruskal-Wallis tests for Observed richness (χ² = 3.65, p = 0.302), Shannon index (χ² = 6.08, p = 0.108), or Simpson index (χ² = 6.05, p = 0.109). These findings were further supported by generalized linear models: the quasi-Poisson GLM for observed richness did not reveal significant effect of site, and the Gaussian GLMs for Shannon and Simpson indices also indicated no statistically significant differences. A consistent, though non-significant, trend toward higher diversity was observed in Site 3 across both Shannon and Simpson indices (Fig. 2). These results suggest a relatively homogeneous distribution of gut eukaryotic alpha diversity across the studied sites.

Figure 2 Alpha diversity measurement.

Observed Richness, Shannon and Simpson diversity index values from eukaryotic composition in fecal bats roosts on average for each site. No statistical differences were found between sites.

Native vegetation percentage is the main factor explaining the eukaryotic gut community structure between sites

To assess differences in community composition across sampling sites, we conducted a permutational multivariate analysis of variance (PERMANOVA) using the Bray–Curtis dissimilarity matrix. Results indicated a significant effect of site on community structure (R² = 0.52, F = 2.90, p < 0.001), indicating that 52% of the variance in community composition was explained by location. Using presence–absence data and Jaccard dissimilarities, a second PERMANOVA also revealed significant differences in community composition among sites (R² = 0.41, F = 1.83, p < 0.001). This indicates that 41% of the variation in species composition was attributable to site. The ‘envfit’ function revealed that environmental variables significantly associated with community composition differed slightly depending on the dissimilarity index used. In NMDS based on Bray–Curtis distances, variables such as natural vegetation (NaV), anthropized cover (AnC), water bodies (Wat), and riparian vegetation (Rip) showed strong associations with the ordination (p < 0.001, p < 0.001, p < 0.001, and p = 0.011, respectively) (Fig. 3). In the Jaccard-based NMDS (Fig. S2), similar variables were significant, but orchards (Orch, p = 0.001), prairies (Pra, p = 0.003), urban areas (Urb, p = 0.002), and water bodies (Wat, p < 0.001) exhibited higher r² values, suggesting that these factors may be more influential in shaping species presence/absence patterns rather than abundance. In contrast, vineyards (Vin), which were significant in the Bray–Curtis ordination (p = 0.019), were only marginally significant under Jaccard (p = 0.057).

Figure 3 Non-metric multidimensional scaling (NMDS) of eukaryotic community composition based on Bray-Curtis dissimilarities across different habitat types.

The arrows names corresponded to Rip, Riparian habitats; NaV, native vegetation; AnC, annual crops; Orch, orchards; Wat, water bodies; Pra, prairie; Ru, rural areas; Urb, urban areas; Hed, hedgerows; Vin, vineyards fields.

Principal component analysis (PCA) was used to summarize variation in site-level environmental variables (Fig. S3). The first two principal components (PC1 and PC2) explained major gradients in landscape composition and were significantly associated with gut eukaryotic community structure. Vector fitting analysis on the PCA ordination revealed that both PC1 (r² = 0.90, p = 0.001) and PC2 (r² = 0.59, p = 0.019) were significantly correlated with community composition. Similarly, PERMANOVA models using PC scores also revealed significant associations with beta diversity, both for Bray-Curtis (PC1 p = 0.001, PC2 p = 0.011) and Jaccard distances (PC1 p = 0.001, PC2 p = 0.001).

In contrast, generalized linear models using PC1 and PC2 as predictors did not reveal significant associations with alpha diversity indices, including Observed Richness (PC1 p = 0.797, PC2 p = 0.577) and Shannon diversity (PC1 p = 0.127, PC2 p = 0.758). Sites clustered in accordance with these environmental gradients, with Site 4 strongly associated with AnC and Vin, and Site 3 with higher NaV values (Fig. S3).

High representation of Apicomplexa and other ecologically relevant eukaryotes in bat fecal samples

For visualization purposes, only ASVs assigned at the phylum level were included. At the phylum level, only Apicomplexa, Ascomycota, and Basidiomycota were present at all the sampling sites. Among the total reads, Apicomplexa was the most abundant, although its relative abundance varied by Site: Site 1 had 86.1% of the total reads assigned to this phylum, followed by Site 2 (75%), Site 3 (37.6%), and Site 4 (16.2%) (Fig. 4).

Figure 4 Relative abundance of dominant eukaryotic phyla in bat fecal samples across sampling sites.

Heatmap showing the percentage of amplicon sequence variants (ASVs) assigned to the most abundant eukaryotic phyla detected at each site. Color intensity ranges from orange (highest abundance) to blue (lowest abundance).

The phylum Ascomycota was particularly abundant at Site 3 (41.9%), and Basidiomycota was more abundant at site 4 (33.5%). Platyhelminthes and Zoopagomycota were primarily detected at Site 3, whereas Euglenozoa were present only at Site 4 (9.9%), and Dinoflagellata were present only at Site 1 (0.7%). On the other hand, Mucoromycota was present at Site 2 (1.1%) and Site 3 (0.4%) (Fig. 4).

At the genus level, Eimeria from the Apicomplexa phylum was the most abundant genus found in our samples (Fig. S4).

Distribution of Apicomplexa across sampling sites

A total of 14 amplicon sequence variants (ASVs) belonging to the Apicomplexa phylum were detected across sites (Fig. 5). These ASVs included members of the genera Eimeria, Cryptosporidium, Cystoisospora, and Adelina, along with several taxonomically unclassified lineages.

Figure 5 Maximum likelihood phylogenetic tree of Apicomplexa ASVs identified in bat gut samples.

The tree was inferred under a GTR + I + Γ model based on aligned ASV sequences. Tip labels include the ASV identifier and, when available, the assigned genus. Colored circles represent the presence of each ASV in different sampling sites, with circle size proportional to the number of sequence reads. Multiple adjacent circles indicate detection of the same ASV in more than one site. Scale bar indicates expected substitutions per site.

The most abundant ASVs corresponded to the genus Eimeria (ASV17 and ASV57), were present in multiple sites, suggesting a relatively wide distribution and possibly a high prevalence among host organisms. In contrast, ASV111 (Cryptosporidium) and ASV109 (Cystoisospora) showed more restricted distributions, appearing only in a subset of sites. This could reflect ecological specialization or lower prevalence within the sampled communities.

A substantial proportion of the detected ASVs were unclassified, indicating either limited representation of these taxa in current reference databases or the presence of potentially novel Apicomplexan lineages. Notably, several unclassified ASVs were found across all sites, underscoring the need for further taxonomic and functional characterization. Of particular interest is ASV129, assigned to the genus Adelina, a group rarely reported in metagenomic studies.

Special attention is drawn to Site 3, this site harbored a broader range of ASVs, including both classified and unclassified taxa, suggesting a more complex parasitic community or a richer diversity of hosts. Conversely, Site 4 showed the lowest diversity, with fewer ASVs detected overall, potentially reflecting a simpler community structure or more constrained ecological conditions.

These results highlight the spatial heterogeneity of Apicomplexan communities across sites, shaped potentially by differences in host availability, environmental factors, or anthropogenic impacts.

Discussion

Our study revealed that phyla Apicomplexa, Ascomycota, and Basidiomycota were consistently present in bat fecal samples, with Apicomplexa being the most abundant.

Comparative studies show important differences in phylum abundance across species and regions. For example, in phytophagous and insectivorous bats from China, Ascomycota was more abundant than Basidiomycota and Apicomplexa (Li et al., 2018), contrasting with our findings where Apicomplexa dominated. Within Ascomycota, we identified genera such as Metschnikowia, Nakaseomyces–Candida clade, Clavispora–Candida clade, Aspergillus, and Aureobasidium. While Candida and Aspergillus include species known to be opportunistic pathogens in humans (Narayanan et al., 2021), their presence in bat feces should be interpreted with caution, because are commonly found in cave environments and bat roosts (Nováková, 2009; Vanderwolf, Malloch & McAlpine, 2016). Also, Aspergillus fumigatus var. fumigatus, Aspergillus sydowii, and Cladosporium cladosporioides, are known to utilize keratinous materials as nutrient sources were isolated from live bats (Voyron et al., 2011), suggesting a functional adaptation to the organic matter associated with bats.

Aureobasidium has been identified in bees (Goulson & Hughes, 2015; Tian et al., 2018), which are key pollinators in agricultural ecosystems, including vineyards (Kratschmer et al., 2018). The presence of these genera in our samples likely reflects the insectivorous diet of M. arescens, which may include hymenopterans, like bees.

To clarify the taxonomic identity of the Apicomplexa detected in bats, we conducted a phylogenetic analysis based on the ASVs identified in this study. This approach contextualized the dominance and broad distribution of Eimeria, as well as the presence of several unclassified lineages found across all sites. While most sequences could not be identified at the genus level, some were classified as Cryptosporidium and Cystoisospora, known zoonotic pathogens affecting humans and animals (Yang et al., 2021; Casmo et al., 2018; Schiller, Webster & Power, 2016). Additionally, species from the genus Adelina, identified at two of the four sites, have been described as parasites of certain insects (Bekircan & Tosun, 2021). These findings enhance our understanding of host specificity, potential pathogenicity, and the ecological diversity of bat-associated Apicomplexa. Notably, several ASVs did not cluster with any known genus, indicating the presence of unclassified or potentially novel Apicomplexan lineages in M. arescens. The detection of these diverse and partially uncharacterized taxa highlights the ecological complexity of gut eukaryotic communities in bats and underscores the need for further molecular and functional studies.

The prevalence of the Eimeria genus, known for including coccidian parasites of mammals, highlights the ecological relevance of this taxon among bat gut eukaryotes. Although Eimeria species are typically host-specific and their pathogenicity in humans has not been demonstrated, some species like E. tenella, E. acervulina, and E. maxima, cause severe diarrhea and high mortality in poultry (Boulton et al., 2018; El-Shazly et al., 2020). Additionally, gastrointestinal illness and mortality have been observed in other mammals such as rabbits and rodents (Peeters et al., 1984), underscoring the impact of these parasites in animal health, which may vary depending on the species and age of the host. Also, molecular evidence of Eimeria in bat feces has been reported in recent studies (Couso-Pérez et al., 2022), supporting the relevance of our findings.

The percentage of native vegetation significantly structured the eukaryotic gut community of M. arescens. This supports growing evidence that gut microbiota reflects environmental conditions, such as habitat degradation and resource availability, with implications for wildlife health and conservation (Heni et al., 2023; Lobato-Bailón et al., 2023). Preserving natural habitats is essential to maintain microbial ecology and the vital ecosystem services provide by bats, such as pollination and insect pest control (Ramírez-Fráncel et al., 2022).

On the other hand, Schwensow et al. (2022) found no effect of habitat on intestinal parasite diversity in small mammals using egg morphology with McMaster flotation technique. In contrast, our use of high-throughput sequencing provides broader range of eukaryotic taxa. Given the methodological differences in sensitivity and taxonomic resolution, comparisons between studies should be made with caution.

Despite our sample size is limited, our results suggest that habitat characteristics may play a more important role in shaping eukaryotic gut community than disturbance per se. In particular, bats from Site 3—characterized by higher native vegetation cover and a predominantly rural context—showed a trend toward greater gut eukaryotic diversity, although the difference was not statistically significant. This observation may reflect the influence of lower anthropogenic pressure in this site and aligns with findings from other studies that emphasize the role of habitat structure and quality in shaping host-associated bacterial and eukaryotic communities (Schwensow et al., 2022; Fackelmann et al., 2021; Barelli et al., 2020). These patterns highlight the importance of considering habitat heterogeneity and ecological context when evaluating the impact of human activities on gut eukaryotic diversity in wildlife.

Similar patterns have been observed in other mammals. A recent study by Chege et al. (2024) in wild baboons, found that protists and fungi are widespread gut residents, and that social group membership strongly influenced eukaryotic composition, whereas age, sex, and season had no detectable effect. While our study focuses on a different host species, the influence of habitat context on gut eukaryotes aligns with these findings and highlights the ecological complexity of these communities.

Previous studies have demonstrated that pristine and undisturbed landscapes promote a more resilient gut microbial communities in bats (Lobato-Bailón et al., 2023), while habitat fragmentation is associated with increased variability, as seen in common vampire bats (Ingala et al., 2019). Similarly, anthropogenic disturbances, not fragmentation alone, affects the gut microbiome in Tome’s spiny rat (Proechimys semispinosus) (Fackelmann et al., 2021). However, these studies focus mostly on bacterial components of the microbiome (Ingala et al., 2019; Huang et al., 2022; Fleischer et al., 2024), with limited focus on the eukaryotic component. Our research addresses this gap, contributing to a broader understanding of how landscape variables shape the eukaryotic gut community in bats, which are integral to their health and ecological roles.

From a One Health perspective, the detection of potential pathogens underscores the complexity of interactions between wildlife, microbes, and environmental change. While we do not claim direct zoonotic risk from the Eimeria sequences identified, these findings support the need to monitor potential pathogenic eukaryotes in wildlife, particularly in landscapes undergoing rapid transformation due to land use change.

While this study provides valuable insights into the gut eukaryotic communities of M. arescens and its association with landscape features, we recognize potential limitations and/or biases which could limit the inferences from our findings. First, information on the sex and age of the sampled bats was not available, which could influence microbiome composition and interpretation. Second, while our sampling design targeted a broad range of colonies and habitat types, the number of samples per site was limited. Each sample included approximately 20 fecal pellets, likely from multiple individuals, but individual-level resolution was not possible. Third, our cross-sectional design does not account for short term temporal variation in microbiota composition. Additionally, while native vegetation was identified as a significant structuring factor, unmeasured environmental or physiological variables may also play a role.

This study represents an initial phase of research in which we plan to increase sample size and address more variables, to help us understand how human activities may impact the habitats and populations of bat species. Our results provide the first novel insights into the understudied gut eukaryotes in a vulnerable endemic bat inhabiting fragmented landscapes. Thus, although species-specific, our insights might be extrapolable to other endemic bats inhabiting landscapes with increasing urbanization and land-use changes such as the ones studied here. Future research should aim to integrate longitudinal sampling and additional environmental variables to build a more comprehensive understanding of these dynamics and to provide managers and practitioners adequate information to design evidence-based conservation strategies.

Conclusions

This study provides the first characterization of the gut eukaryotic community in the endemic Chilean bat Myotis arescens, revealing a consistent dominance of Apicomplexa—particularly Eimeria—across sampling sites. The high abundance of this phylum underscores its ecological relevance in bat gut communities and highlights the need to preserve balanced host–parasite dynamics in the face of environmental change. This study confirmed that native vegetation significantly structures the gut eukaryotic communities of M. arescens, supporting its potential use as a bioindicator for habitat quality. Differences in beta diversity between sites suggest that landscape composition influences gut eukaryotic community structure, even in the absence of statistically significant variation in alpha diversity.

From a One Health perspective, our findings emphasize the value of monitoring eukaryotic wildlife communities—especially in rapidly changing habitats—to better understand host–microbe dynamics and inform conservation strategies aimed at reducing spillover risks.

Our findings contribute to understanding the gut eukaryotes communities in bats, an underexplored area, particularly in countries of the Global South, and emphasize the ecological implications of habitat loss. Limitations such as the lack of age and sex data, limited sample size per site, and the cross-sectional design should be considered when interpreting the results. Future research should address short term temporal variation and individual-level resolution.

Conservation strategies that restore native vegetation and promote habitat connectivity are critical for preserving bat health, mitigating zoonotic risks, and maintaining their ecological roles as pollinators and pest controllers.

Supplemental Information

Supplemental Information 1 Rarefaction curves of identified microbial eukaryotic ASVs.

Supplemental Information 2 Non-metric multidimensional scaling (NMDS) of eukaryotic community composition based on Jaccard distance.

Supplemental Information 3 Biplot of the principal component analysis (PCA) of site-level environmental characteristics.

Points represent sampling sites, colored by site ID. Arrows indicate the direction and strength of environmental variables contributing to the ordination space. Sites 2 and 4 are characterized by high levels of anthropogenic cover (AnC), vineyards (Vin), and riparian vegetation (Rip), while Site 3 aligns more with natural vegetation (NaV) and rural areas (Ru).

Supplemental Information 4 Heatmap of top most abundant ASVs.

Heatmap of relative abundance of the 30 most abundant ASV from each site. The numbers indicate the percentage of ASV abundance in each sample site. Orange represents the highest percentage and blue the lower percentage.

Additional Information and Declarations

Competing Interests

The authors declare that they have no competing interests.

Author Contributions

Lia Ramírez-Fernández analyzed the data, prepared figures and/or tables, authored or reviewed drafts of the article, and approved the final draft.

Mónica Saldarriaga-Córdoba conceived and designed the experiments, performed the experiments, authored or reviewed drafts of the article, and approved the final draft.

Andrea X. Silva performed the experiments, authored or reviewed drafts of the article, and approved the final draft.

Constanza Napolitano analyzed the data, authored or reviewed drafts of the article, and approved the final draft.

Annia Rodríguez-San Pedro conceived and designed the experiments, performed the experiments, analyzed the data, prepared figures and/or tables, authored or reviewed drafts of the article, and approved the final draft.

Field Study Permissions

The following information was supplied relating to field study approvals (i.e., approving body and any reference numbers):

Fieldwork was approved by the Comisión Nacional de Investigación Científica y Tecnológica (CONICYT) under the sponsorship of the University of Chile (project number: 3160188).

Data Availability

The following information was supplied regarding data availability:

The 18S metabarcoding dataset is available at the European Nucleotide Archive (ENA): PRJEB78431.

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
