# Peer review of "Eukaryotic gut community of the bat Myotis arescens in anthropized landscapes in Chile"

_PeerJ, doi:10.7717/peerj.19563_

## Round 0.1 · original submission · Major Revisions

Please provide a detailed point-by-point rebuttal letter to each of the reviewers' comments, along with your revised manuscript.

·

Basic reporting

Minor comments:
1) Line 163: “18S rRNA gen.” should be “18S rRNA gene.”
2) Line 177: “the Silva v.132 database.”, please add a reference here.
3) Line 230 “Eimeria” should be italic.

Experimental design

1) “After excluding samples with fewer than 10 reads to ensure reliable comparisons, ……”. Based on my understanding, the authors removed those samples with fewer than 10 reads (i.e. three samples: 13, 41 and 43). They did as shown in Shannon diversity (Figure 2). However, for beta diversity, all samples (3 in each site) seemed to be included as shown in Figure 3. I suggest authors to explain this inconsistency here.

In addition, after removing those samples with fewer than 10 reads, only one same (43) was retained in site 4. I wonder whether this will have an effect on comparison results between sites?

2) “In total, 71,995 Illumina sequence reads were obtained from 12 samples, 22,481 of which remained after quality filtering.” Please add specific quality filtering procedures in Materials and methods.

3) How about the sexes and ages of bats used in this study? The authors should state this information in the sampling section or say something in the Discussion section as one of limitations of this study ?

Validity of the findings

no comment.

Additional comments

Ramírez-Fernández et al. has assessed the impacts of landscape anthropization on the eukaryotic gut microbiome in a small insectivorous bat species endemic to Chile (Myotis arescens) and they also identified some potential pathogenic eukaryotes. Although bats are natural reservoirs of various zoonotic pathogens, most of studies have focused on viruses. Thus, this study highlights an importance of more studies on non-virus pathogens in bats in future, in particular for bats from urban regions where human and bats are in close connects.

Reviewer 2 ·

Basic reporting

I think the work by Lia Ramirez-Fernandez is timely and interesting. I have no major qualms with the work. Here and there the message and clarity could be improved.

I have a list of minor comments:

Anthropized and anthropization are the correct terms but I would recommend to the authors to choose synonyms that are more commonly searched after. For instance, “land use change” or “landscape change” are much more commonly used and thus your work will be easier to find in scholarly searches. E.g. your very known review reference in L76 is never once using the word anthropization

Overall, I think the author have a terrific command over the English language. Just in some occasions, I recommend the authors to consider having a native English speaker read over their work to make the language easier and more accessible. This could solve clunky sentences. For example: “Landscape anthropization, the transformation of natural landscapes into human-modified environments, significantly alters local ecosystems, affecting habitat structure, biodiversity, and ecosystem functioning” could be written as: “Land use change is the transformation of pristine ecosystems to human dominated landscapes at the expense of habitat structure, biodiversity and ecosystem functions.”
In the sentence: “Bats, as key…” there is a logical leap. If bats are so vulnerable to anthropization then how does anthropization increase their zoonotic potential because interactions should be reduced, right? Rephrase to make clearer what you mean.
When you report results offer the reader a direction. When I read about your findings of an influence of vegetation on the eukaryotic gut community, I cannot decipher whether the influence is positive or negative based on your writing.

Í would prefer if you universally use “eukaryotic gut community” or “gut eukaryotes” instead of sometimes “eukaryotic gut microbiome” and sometimes “gut microbiota community” – it confuses the reader, because most studies on the gut microbiome will be focused on bacteria.

L63: delete development – and is it important that those are local ecosystems or is it more important that they are pristine or wild or natural…

L70: Roosting sites is a bat specific term; since this is still your general introduction use a term more universally describing “resting sites”…

L71: Here your rationale is on point; this is missing from the abstract

L78: instead of citing an old review here I would encourage the authors to cite primary literature that found interesting pathogens in bats from more recently (avoid MDPI journals)

L81: There is actually no direct contact with humans in the Eby et al. 2022. Hendra is transmitted to horses from fruit bats, and then to humans. That’s indirect. Direct contract would be in the case of corona-virus positive bats defecating into water sources (Meyer et al. 2024) or eating meat of ebola-infected bats (Leroy et al. 2009).

L92: “In Chile, despite the vital ecological roles that bats play (cite a reference and delete the potential implications bit), few studies… - we have to be quite careful about how we communicate bat borne disease and bats don’t have immediate public health implications (if we don’t encroach on their habitat) see Plowright et al. 2024 Nat Comms and Weber et al. 2023 Proc B

L97: Do we know anything about their lifespan. I am just curious.

L100-103: rephrase that sentence

L109: cite a few of those studies that are only reporting on the gut bacterial community in bats.

L119: I would give the locations you sampled names. Those names should be included in the map and then used in the figures instead of 1-4 because having names is easier to remember than 1,2,3, and 4 stands for.

L126: cite the Köppen climate classification

L134: spelling – roost

What is the rational behind so few samples when there should be plenty of fresh samples daily in a colony of 40-100 animals – was this purposefully treated as a pilot study and what will you plan to do in the future.

L177: Why Silva 132 – there is a more current database?

L180: Besides Shannon, I would be interested to see absolute species richness too. Could be a supplementary plot

L182: I take it that KW was used because Shannon wasn’t normally distributed? Alternative and in addition, when you analyse species richness you could use a glm with Poisson distribution.

L187: Like with Shannon, which considers abundance of taxa, Bray also weighs samples based on the relative abundance of reads. I would be keen to also see it based on Jaccard distance, which is qualitative.

L202: which sample was excluded because in your NMDS plot you still have 12?

L206: give the site a name. Avoid ; instead write clear sentence. Report p-values only to the third decimal place

L210: write: The NMDS analysis revealed that the samples clustered by location (ANOSIM; R = 0.7; p < 0.001; Fig 3)

L213: that to also have NaV (and other site characteristics) in your model is missing from the method section

What you could do is merge the site characteristics based on a PCA analysis (see Schwensow et al. 2022 JAE) – that paper is actually an interesting comparison to your work because it does not see a clear effect of habitat characteristics directly influencing eukaryotic parasites albeit affecting the gut bacterial community

L224: site 6? Your plots only have 4 sites

L220 and following: This is maybe the biggest ask but I think it would be a great addition to provide a phylogenetic tree of the ASVs found in the apicomplexa phylum and compare it to known bat apicomplexans (and some of other animals possibly) because it allows the reader to better visualise what you found. A phylogenetic tree should also showcase which group of apicomplexans are human, livestock and wildlife pathogens because we then can immediately assess pathogenic potential of the apicomplexans found in your samples.

L247: Like your findings the Lobato-Bailon study is based on very few samples. I think its worth discussing your limitations (low sample size) and that of this study in the discussion briefly. By contrast, work that includes the Schwensow paper but also from Fackelmann et al. 2021 (which you cite) show very nicely and statistically sound that habitat differences shape microbiomes (although this was done on rodents). One has to be also open to the possibility that its not disturbance per se but differences in habitats (as your results suggest) and this is something that has been found for other hosts (e.g., Barelli et al. 2020 msystems)

L249: cite also Fleischer et al. 2024 Sci Total Environ here too

L254: There is a preprint out on eukaryotic composition of baboons (Chege et al. 2024) – worth discussing?

L266: I think you can invest one or two more sentence on that recent finding in bats. What host? How common? Any habitat effects? Etc This is where a phylogeny would come in handy because you could compare your sequences to theirs.

L282-287: does it though? I would wish for a more careful discussion instead of making bats the bad boys.

L294: It was not clear in the methods that the 3 samples were pooled samples. Please make that clear. Given your future plans, I would encourage the authors to use individual pallets rather than pooled samples.

L297 what do you mean with temporal variability – short term (e.g., Schmid et al. 2023 Funct Ecol) or long-term (e.g. Sadoughi et al. 2022 Microbiome) ?

Experimental design

This is fine. Although sometimes clarity in the method section is missing (e.g., statistics and sample sizes; see comments)

Validity of the findings

The findings are statistically as robust as they can be given that they are based on a small sample size. I would encourage the authors that in future work they substantially expand sampling and even in pilot studies, sample sizes should be >10.

Additional comments

none

·

Basic reporting

The study by Ramírez-Fernández et al. provides valuable insights into bat microbiome analysis, particularly highlighting how habitat loss can impact various species, with a specific focus on a insectivorous bat. After carefully reading the MS, I am attaching my comments, and my suggestion is major revision.
Title: Why do you want to mention ‘Microbial eukaryotes’?
Abstract
Line 30-32: It remains unclear whether wildlife pose a threat or if they can become ill or be affected by different pathogens.
Line 32: Why key ecological players?
Line 52? Why? There are species of Eimeria isolated from bats that can be zoonotic? If not I believe this phrase is too strong.
Line 55: Is unclear how they can serve as bioindicators.
Line 56: Again, the same problem with zoonotic Eimeria.
Introduction
Line 63: Put reference. Who said this? What studies have shown this?
Line 63- 65: Put references for each statement, not just at the end.
Line 72: More references.
Line 73-75: Put references for each statement, not just at the end.
Line 78: More references.
Line 80: There is no study on this??
Line 85: I do not agree with this statement. Nowadays there are studies.
Line 92: Maybe you should mention the number of bat species present in the country.

Experimental design

Materials and methods
Line 152: How did you determine the freshness of feces?
Line 153: What does low temperatures mean?

Validity of the findings

It is not sufficient described the results, especially the Eimeria part.

Discussion
What I miss in this part, and it is highly important is the discussion of your results compared with other studies.
Line 240-241: What species of Eimeria isolated from bats are zoonotic? There is no reference for this.
Line 263: The same problem regarding the species that can cause severe diarrhea. Not all are the same and the impact on the health status of animals is different in function of the age of the animals.
Line 265: Reference?
Line 267: Put more details how these two studies prove the statement that bat Eimeria species are pathogenic and can pose zoonotic problems.
Line 269-280: What else cand your results mean? Only the type of diet they consume?
Line 282-284: How? It is mentioned that Eimeria spp., no species that can have zoonotic potential. And this is the first time you mention Candida spp. and Aspergillus spp.
Line 314: I do not agree with this statement.

Figure 2: there are no statistical differences?

---

## Round 0.2 · accepted · Accept

Your paper is now accepted.

·

Basic reporting

No further comments.

Experimental design

No further comments.

Validity of the findings

No further comments.

Additional comments

All of my former concerns have been responded correctly. No further comments for this new version of the MS.

Reviewer 2 ·

Basic reporting

I have seen this manuscript now for the second time and it has substantially improved in clarity and concision.

While I do not agree with not rarefying prior to beta diversity analysis (see Schloss et al. 2024), the authors are aware of the limitations of their already low sample size and communicate this openly.

I also do not agree to keep unclassified phyla to improve read coverage but it does not seem to affect the results from the previous to this version much. Again, with this sample size the authors will not shake up our understanding of the importance of eukaryotic microbiota, but rather give an indication whether its worth persuing in the future (pilot).

Experimental design

fine.

Validity of the findings

they have been very careful in how the findings are now described and put into context.